DATA RELEASE

# Distribution of mosquitoes (Diptera: Culicidae) in Thailand: a dataset

Chutipong Sukkanon[1], Wannapa Suwonkerd[2], Kanutcharee Thanispong[2], Manop Saeung[3], Pairpailin Jhaiaun[3], Suntorn Pimnon[3], Kanaphot Thongkhao[4], Sylvie Manguin[5] and Theeraphap Chareonviriyaphap[3,6,*]

1 Department of Medical Technology, School of Allied Health Sciences, Walailak University, Nakhon Si Thammarat, 80160, Thailand
2 Division of Vector Borne Diseases, Department of Disease Control, Ministry of Public Health, Nonthaburi, 11000, Thailand
3 Faculty of Agriculture, Kasetsart University, 10900, Bangkok, Thailand
4 Office of Disease Prevention and Control Region 11, Nakhon Si Thammarat, Thailand
5 HSM, University of Montpellier, CNRS, IRD, 34093, Montpellier, France
6 Royal Society of Thailand, 10300, Bangkok, Thailand

## ABSTRACT

Mosquitoes play a crucial role as primary vectors for various infectious diseases in Thailand. Therefore, accurate distribution information is vital for effectively combating and better controlling mosquito-borne diseases. Here, we present a curated dataset of the mosquito distribution in Thailand comprising 12,278 records of at least 117 mosquito species (Diptera: Culicidae). The main genera included in the dataset are *Aedes*, *Anopheles*, *Armigeres*, *Culex*, and *Mansonia*. From 2007 to 2023, data were collected through routine mosquito surveillance and research projects from 1,725 locations across 66 (out of 77) Thai provinces. The majority of the data were extracted from a Thai database of the Thailand Malaria Elimination Program. To facilitate broader access to mosquito-related data and support further exploration of the Thai mosquito fauna, the data were translated into English. Our dataset has been published in the Global Biodiversity Information Facility, making it available for researchers worldwide.

**Submitted:** 30 June 2023

\* Corresponding author. E-mail: faasthc@ku.ac.th

Preprint submitted at https://doi.org/10.5281/zenodo.8231848

Included in the series: *Vectors of human disease series* (https://doi.org/10.46471/GIGABYTE_SERIES_0002)

**Subjects** Ecology, Biodiversity, Zoology

## DATA DESCRIPTION

### Context

The prevalence of mosquito-borne diseases remains a significant public health challenge in Thailand. Among such diseases, the most notable ones are malaria, dengue fever, chikungunya, and Zika. These diseases significantly impact both rural and urban populations, resulting in substantial morbidity and mortality rates [1]. Accurate and up-to-date information on mosquito distribution plays a vital role in fighting these diseases. Consequently, having access to accurate information regarding mosquito distribution throughout the country is of utmost importance for implementing evidence-based vector control strategies. By understanding the spatial distribution of mosquitoes, authorities can effectively target areas at higher risk and deploy appropriate preventive measures to mitigate the impact of these diseases on public health [2].

To construct a dataset of mosquito distribution in Thailand, a comprehensive and meticulous review of information was conducted. No specific temporal range limitations were imposed to include the maximum amount of available data. Most of the data presented in this dataset has been extracted from the Thailand Malaria Elimination Program database [3] by the Division of Vector Borne Diseases, Department of Disease Control, Ministry of Public Health, Thailand. While the Thailand Malaria Elimination Program database has undergone digitization, it is important to note that the information was deposited in the Thai language only, thus restricting access to a wider audience. Therefore, our objective was to address and overcome this limitation by expanding the accessibility of mosquito-related data, thus supporting further exploration of Thailand's mosquito fauna from a broader audience. The data were also curated from various scientific studies carried out by the Department of Entomology, Faculty of Agriculture, Kasetsart University (KU), and the Department of Medical Technology, School of Allied Health Sciences, Walailak University.

For each recode of mosquito occurrence, our dataset includes fields describing their: (i) taxonomy (scientific Name, kingdom, phylum, class, order, family, genus, specific epithet, scientific Name Authorship, taxonRank); (ii) collection details (event ID, occurrence ID, event date, sampling protocol); (iii) geolocation data (county, country Code, locality, location ID, decimal Latitude, decimal Longitude, geodetic Datum). The temporal coverage of the database spans from February 19, 2007, to April 4, 2023, encompassing a significant timeframe. The dataset contains 12,278 records of at least 117 mosquito species, mainly of the genera *Aedes*, *Anopheles*, *Armigeres*, *Culex*, and *Mansonia*. Also, the mosquitos were collected from 1,725 locations in 66 provinces out of 77 across Thailand. The data are provided in the Darwin Core format [4]. Our data are published in the Integrated Publishing Toolkit of the Global Biodiversity Information Facility (GBIF) [5] and are publicly available for use by others (https://doi.org/10.15468/tbd7fz) [6].

## METHODS

### Larval collection

As part of 'baseline surveys', the collection of mosquito immature stages aimed to gather comprehensive information on the distribution of vector species in a specific area. This involved exploring and identifying larval habitats and seasonal patterns to understand mosquito populations more broadly. Unfortunately, the data from the field studies lacked specific information about the larval habitat. Both larvae and pupae were collected using the dipping technique and/or water-holding container inspection. A thorough inspection was conducted on all indoor and outdoor artificial water containers within a household, as well as other containers within 10–20 m from the house. When mosquito larvae or pupae were detected, they were carefully transferred into a collection bottle using a Pasteur pipette. All immatures were transferred to climate-controlled insectary for further rearing into the adult stage. Adults were then identified using well-established morphological keys [7, 8].

### Mosquito surveillance projects

At Kasetsart University (KU), novel methods for collecting outdoor mosquitoes were investigated to assess and compare alternative techniques for vector surveillance. In the Light Trap project, four light-emitting diodes (blue, green, yellow, and red) and two

fluorescent lights (ultraviolet (UV) and white) were used in the commercial mosquito traps for collecting specimens in urban areas and field sites of Bangkok [9] and Kanchanaburi [10] provinces, respectively. Generally, the mosquito traps were equipped with an electrical fan and a UV light source powered by an alternating current of 220–240 V. Then, six traps were used to compare the effectiveness of light traps equipped with different bulbs across the wavelength spectrum. A Latin square experimental design was employed to set up six light traps, each equipped with different lights. These traps were systematically rotated among six designated locations. Traps were then operated simultaneously from 6 p.m. to 6 a.m. Captured mosquitoes were removed at 3-hour intervals. During 36 collection nights, six replications were conducted for each location, totaling 216 trap nights. At urban sites in Bangkok, the mosquito collection was conducted between February and August 2019, once every month. At Kanchanaburi, field trapping was performed on six consecutive nights, every two months, across the dry (April), wet (June and August), and cold (October and December) seasons of 2020. Collected mosquitoes were transferred into labeled holding cups using entomological mouth aspirators and brought to the laboratory for further analyses. Sex and genus/species were separated under a dissecting stereo microscope (ST-6 Zoom Stereo Microscope, Scilution Co., Ltd., Bangkok, Thailand). Mosquitoes were killed in the freezer (−20 °C) and then morphologically identified using well-established morphological keys [7, 8, 11–14]. All primary *Anopheles* species, including the *Anopheles minimus* complex, the *Anopheles maculatus* group, and the *Anopheles dirus* complex, were then further identified using molecular identification techniques [15–18].

As part of 'mosquito surveillance projects', the effectiveness of the 'gold standard' outdoor human landing collection (OHLC) was also compared with alternative mosquito collection methods, namely human double net trap (HDNT), human decoy trap (HDT), and UV light trap (UVLT). This study aimed to minimize reliance on human-based mosquito collection methods and decrease the potential risk of mosquito bites on human volunteers. The study was conducted in Sai Yok District, Kanchanaburi Province. The HDNT employed a dual-net system comprising an inner and outer net. The inner net thoroughly protected the volunteer acting as bait and sleeping on a mattress. The outer net, stitched to the inner net, was elevated 30 cm above the ground. Every hour attracted mosquitoes were collected by another volunteer using a flashlight and a mouth aspirator, and the collection took 10 min. The HDNT collections were carried out by volunteers overnight, from 6 p.m. to 6 a.m., under regular supervision. For the HDT approach, the circumference of the black thermally heated trap was covered with a transparent adhesive plastic sheet. A protected volunteer slept in a tent, positioned approximately 5 m from a volunteer, as a source of odor. The odor was funneled down a 6 m PVC pipe towards the trap. At the end of each collection period (6 p.m. to 6 a.m.), the adhesive sheets were removed and transported to the laboratory. Collected mosquitoes were then removed using forceps. The OHLC was performed according to the standard guidelines [19]. Finally, a commercial mosquito trap (Black Hole™ Mosquito Trap, Bio-Trap Inc., Seoul, Korea) equipped with UV light was used for the UVLT approach. In a pre-assigned Latin square design, the four outdoor collection methods were rotated nightly among three predetermined trapping positions. Captured mosquitoes were then identified using well-established morphological keys [7, 8, 11–14].

## Thailand Malaria Elimination Program dataset

Most of the adult mosquito data were extracted from the public domain of the Thailand Malaria Elimination Program database [3]. Data were deposited and digitized only in Thai.

Briefly, the data was exported into Microsoft Excel files, and all Thai information (such as subdistrict and village names) was then transliterated using either The Royal Institute of Thailand's Transliterated Words Database System [20] or the thai2english database [21]. Mosquitoes were collected as part of routine mosquito surveys of the Thailand Malaria Elimination Program using the OHLC technique across Thailand. Both indoor and outdoor collections were performed by experienced researchers and skilled public health professionals. Mosquitoes were collected using a flashlight and a mouth aspirator every hour from 6 p.m. to 6 a.m., and each collection required 50 min. There was a resting period of 10 min between each collection. Collectors were assigned to perform OHLC for a total of 6 h. Two nights of collection were conducted at each location. However, the collected mosquitoes were primarily identified using adult morphological keys. No molecular analysis was performed as the essential tools and reagents required to perform molecular testing were unavailable at each Office of Disease Prevention and Control. Unfortunately, geographic coordinates were not provided for the entirety of the data from the public domain of the Thailand Malaria Elimination Program database. The village/sub-district name listed in the original dataset file was then utilized along with administrative division details to establish a precise location by leveraging Google Earth.

### Data georeferencing process

The World Geodetic System 1984 served as the standard reference for determining the precise geographic coordinates (latitude and longitude) associated with any given location in the dataset. In case geographic coordinates were not provided, as mentioned above, the village/sub-district name listed in the original dataset file was utilized along with administrative division details to establish a representative location by leveraging Google Earth. As a result, these coordinates do not provide precise but representative geographic locations within the corresponding village/sub-district (Figure 1).

### DATA VALIDATION AND QUALITY CONTROL

For the Minimus and Dirus complexes and the Maculatus Group, mosquitoes were identified by experienced taxonomists using well-established morphological keys [7, 8, 11–14] and molecular techniques [15–18]. Data validation was thoroughly screened using the Integrated Publishing Toolkit of GBIF [5]. This toolkit, which utilizes the GBIF software, ensures data validation through its network. Additionally, metadata fields are accessible on the online pages [6].

### RE-USE POTENTIAL

The dataset presented here offers valuable insights into the distribution and identification of mosquito species in Thailand. Researchers and public health professionals can leverage this dataset to enhance their understanding of mosquito-borne diseases, identify vector species, and assess potential transmission risks across various regions of Thailand. The information contained in the dataset can serve as a resource for studies focused on vector control strategies, disease surveillance, and the broader comprehension of mosquito ecology in Thailand.

### DATA AVAILABILITY

The data supporting this article are published through the Integrated Publishing Toolkit of GBIF [5] and are available under a CC0 waiver from GBIF [6].

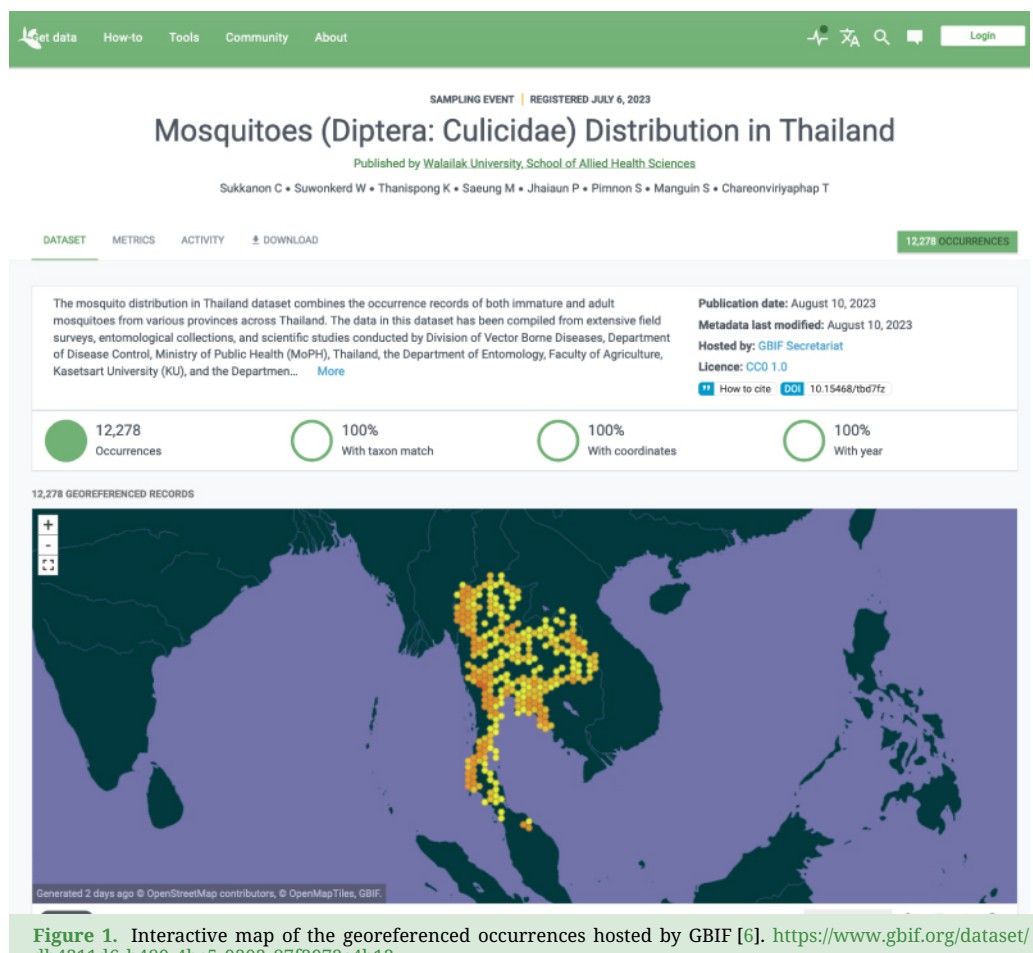

**Figure 1.** Interactive map of the georeferenced occurrences hosted by GBIF [6]. https://www.gbif.org/dataset/db4211d6-b480-4bc5-9202-87f2072c4b12

## ABBREVIATIONS

GBIF, Global Biodiversity Information Facility; HDNT, human double net trap; HDT, human decoy trap; KU, Kasetsart University; OHLC, outdoor human landing collection; UV, ultraviolet; UVLT, UV light trap.

## DECLARATIONS

### Ethics approval and consent to participate

Not applicable.

### Consent for publication

Not applicable.

### Competing Interests

The author(s) declare that they have no competing interests.

## Authors' contributions

CS: conceptualization, validation, data curation, writing - original draft preparation, writing - review & editing; WS: investigation, validation, resources, writing - review & editing; KT: investigation, validation, resources; MS: investigation, validation, resources; PJ: investigation, validation, resources; SP: investigation, validation, resources; KT: investigation, validation, resources; SM: conceptualization, supervision, validation, writing - original draft preparation, writing - review & editing; TC: conceptualization, supervision, validation, resources, writing - original draft preparation, writing - review & editing.

## Funding

This work was partially supported by the Kasetsart University Research and Development Institute (KURDI) Fundamental Fund program [FF (KU) 14.64].

## Acknowledgements

The authors would like to thank everybody who contributed to the creation of these datasets and paper, including the Division of Vector Borne Diseases, Department of Disease Control, Ministry of Public Health, Thailand, the Department of Entomology, Faculty of Agriculture, Kasetsart University, and the Department of Medical Technology, School of Allied Health Sciences, Walailak University. Special gratitude to Paloma Helena Fernandes Shimabukuro and Melissa Liu for their help in the suggestion of dataset and manuscript preparation.

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
