## [Editor Report]

Editor’s AssessmentThere’s a shortage of disease vector data available from Asia, and this data release presents a curated biological collection from across Thailand extracted and made publicly available under a CC0 waiver from Thai database of the Thailand Malaria Elimination Program. Comprising of 12,278 records of at least 117 mosquito species (Diptera: Culicidae), mainly of the genera Aedes, Anopheles , Armigeres, Culex, and Mansonia. During review some more contextual information on how the data was collected was provided. The information now contained here hopefully serving as a resource for studies focused on vector control strategies, disease surveillance, and the broader comprehension of mosquito ecology in Thailand.

---

## [Reviewer Report]

Upload additional filesDRR-202306-01/form/DRR-202306-10_Data-Review-MAT.docxReviewer name and names of any other individual's who aided in reviewer Mary Ann TuliDo you understand and agree to our policy of having open and named reviews, and having your review included with the published papers. (If no, please inform the editor that you cannot review this manuscript.)YesIs the language of sufficient quality?YesPlease add additional comments on language quality to clarify if needed
n/aAre all data available and do they match the descriptions in the paper? YesAdditional CommentsAre the data and metadata consistent with relevant minimum information or reporting standards? See GigaDB checklists for examples <a href="http://gigadb.org/site/guide" target="_blank">http://gigadb.org/site/guide</a>YesAdditional CommentsIs the data acquisition clear, complete and methodologically sound?YesAdditional CommentsIs there sufficient detail in the methods and data-processing steps to allow reproduction?YesAdditional CommentsIs there sufficient data validation and statistical analyses of data quality? YesAdditional CommentsIs the validation suitable for this type of data?YesAdditional CommentsIs there sufficient information for others to reuse this dataset or integrate it with other data?YesAdditional CommentsAny Additional Overall Comments to the AuthorRecommendationAccept

---

## [Reviewer Report]

Upload additional filesDRR-202306-01/form/gx-DR-1688651477_hq.pdfReviewer name and names of any other individual's who aided in reviewer Yeo HuiqingDo you understand and agree to our policy of having open and named reviews, and having your review included with the published papers. (If no, please inform the editor that you cannot review this manuscript.)YesIs the language of sufficient quality?YesPlease add additional comments on language quality to clarify if needed
Are all data available and do they match the descriptions in the paper? YesAdditional CommentsAre the data and metadata consistent with relevant minimum information or reporting standards? See GigaDB checklists for examples <a href="http://gigadb.org/site/guide" target="_blank">http://gigadb.org/site/guide</a>YesAdditional CommentsIs the data acquisition clear, complete and methodologically sound?YesAdditional CommentsIs there sufficient detail in the methods and data-processing steps to allow reproduction?YesAdditional CommentsIs there sufficient data validation and statistical analyses of data quality? YesAdditional CommentsIs the validation suitable for this type of data?YesAdditional CommentsIs there sufficient information for others to reuse this dataset or integrate it with other data?YesAdditional CommentsAny Additional Overall Comments to the AuthorThe dataset has been published under gbif and is already publicly available. Metadata of the records were also provided, including GPS coordinates which will be useful in distributional studies. The records are extensive, covering most of Thailand, and is a valuable resourse for understand vector distribution, dispersal and related work.  Overall the authors have done a good job in describing the various sampling regimes used in this big dataset, including the use of molecular techniques to identify and verify morphologically similar species. Individuals which were unable to be idenified to species level were recorded as genus instead which demonstrated careful verification of the dataset.   I have several suggestions for improving the dataset: 1) It would be good if details of the larval habitats (for records obtained from larval sampling) were provided as well, and 2) if it is possible, it would be useful to indicate which records in the dataset had GPS coordinates inferred from representative locations within the corresponding village/sub-district to differentiate it from records with precise locality information. Please also see attached pdf for minor comments and wording suggestions in the attached pdf.RecommendationMinor Revision